# Phytochemical Profiling and Antioxidant Activities of the Most Favored Ready-to-Use Thai Curries, Pad-Ka-Proa (Spicy Basil Leaves) and Massaman

**DOI:** 10.3390/foods13040582

**Published:** 2024-02-14

**Authors:** Sunisa Siripongvutikorn, Kanyamanee Pumethakul, Chutha Takahashi Yupanqui, Vatcharee Seechamnanturakit, Preeyabhorn Detarun, Tanyarath Utaipan, Nualpun Sirinupong, Worrapanit Chansuwan, Thawien Wittaya, Rajnibhas Sukeaw Samakradhamrongthai

**Affiliations:** 1Centre of Excellence in Functional Foods and Gastronomy, Faculty of Agro-Industry, Prince of Songkla University, Hat Yai 90110, Songkhla, Thailand; kanyamanee.p@psu.ac.th (K.P.); chutha.s@psu.ac.th (C.T.Y.); vatcharee.s@psu.ac.th (V.S.); preeya.h@psu.ac.th (P.D.); nualpun.s@psu.ac.th (N.S.); worapanit.c@psu.ac.th (W.C.); 2Department of Science, Faculty of Science and Technology, Pattani Campus, Prince of Songkla University, Muang, Rusamilae 94000, Pattani, Thailand; tanyarath.u@psu.ac.th; 3Center of Excellence in Bio-Based Materials and Packaging Innovation, Faculty of Agro-Industry, Prince of Songkla University, Hat Yai 90110, Songkhla, Thailand; thawean.b@psu.ac.th; 4Division of Product Development Technology, Faculty of Agro-Industry, Chiang Mai University, Chiang Mai 50200, Changwat, Thailand; rajnibhas.s@cmu.ac.th

**Keywords:** Massaman, spicy basil leaves, curry, characterization of polyphenols, antioxidant activities

## Abstract

Food is one of the factors with the highest impact on human health. Today, attention is paid not only to food properties such as energy provision and palatability but also to functional aspects including phytochemical, antioxidant properties, etc. Massaman and spicy basil leaf curries are famous Thai food dishes with a good harmony of flavor and taste, derived from multiple herbs and spices, including galangal rhizomes, chili pods, garlic bulbs, peppers, shallots, and coriander seeds, that provide an array of health benefits. The characterization of phytochemicals detected by LC-ESI-QTOF-MS/MS identified 99 components (Masaman) and 62 components (spicy basil leaf curry) such as quininic acid, hydroxycinnamic acid, luteolin, kaempferol, catechin, eugenol, betulinic acid, and gingerol. The cynaroside and luteolin-7-O-glucoside found in spicy basil leaf curry play a key role in antioxidant activities and were found at a significantly higher concentration than in Massaman curry. Phenolic and flavonoid compounds generally exhibit a bitter and astringent taste, but all the panelists scored both curries higher than 7 out of 9, confirming their acceptable flavor. Results suggest that the Massaman and spicy basil leaves contain various phytochemicals at different levels and may be further used as functional ingredients and nutraceutical products.

## 1. Introduction

Recently, not only increasing pollution caused by industrial development but also lifestyle, eating and exercising habits, workload, and less relaxing life conditions have been impacting human health in various ways [1], leading to increased stress and the overproduction of free radicals. Unbalanced free radicles produced in the body as a result of stress, anabolism, and catabolism cause macromolecular changes in proteins, fats, carbohydrates, and DNA, increasing the risk of non-communicable diseases and conditions such as stroke, high blood pressure, diabetes, and cancer [2]. Therefore, maintaining balanced food compositions with high levels of antioxidants is beneficial for general wellness. Thai curries and related products, such as fried curry dumplings, steamed buns, and instant stir-fried curries, are an integral part of the historical, cultural, and ethnic background of local Thais and those who have Chinese, Indonesian, and Indian descent. Historically, Thais have used herbs and spices and even curry paste containing galangal rhizomes, chili pods, garlic bulbs, peppers, shallots, and coriander seeds, with health benefits linked to their anti-aging, anti-inflammation, anti-cancer, and antioxidation properties [3,4,5], to treat or relieve common complaints such as stomachache, flu, and acne, following ancestral traditions [6]. Scientific data have shown that phenolic compounds, especially ferulic acid and flavonoids, are antioxidant agents [7,8] that inhibit reactive oxygen species (ROS) and related molecules such as nitric oxide, nitric oxide synthase, and xanthine oxidase, as well as toxic agents produced from free radicals [9]. Allicin derived from crushed or damaged garlic has anti-cancer properties [10], while eugenol plays a key role in antioxidant and antimicrobial activity [11], and the procyanidin oligomer obtained from cinnamon exhibits anti-diabetes properties [12].

Thailand, known as “The Land of Smiles”, is globally renowned for its mouth-watering culinary dishes [6], with Massaman curry regularly recognized as one of the most delicious dishes in the world [13]. Massaman curry combines the sweetness, saltiness, and creaminess of coconut milk with the slight sourness of roasted groundnuts and the distinctive aroma of more than 15 types of herbs and spices such as shallot, dried finger chili, galangal, cumin, lemongrass, clove, kefir lime fruit, cinnamon bark, coriander root, cardamom, and the turmeric rhizome. Massaman curry paste is claimed to be the richest combination of raw materials [14]. Four large Thai companies (Mae-Ploy, Namjai, Aroy-D, and Maesri) produce and export seasoning and famous recipe blends such as green curry, red curry, Tom-Yum, Pad-Thai, and Massaman. Thailand is the second largest global curry paste exporter, following India [15]. 

Spicy basil or Pad-ka-proa is consumed throughout the country by both Thais and tourists. This dish can be cooked quickly, within 5 min, and is combined with various hot spices to suit people of all ages, from children to adults. The taste is mildly salty, sweet, and hot, with the aromas of garlic, pepper, and, particularly, holy basil leaves, which are also used for the treatment of various conditions in Ayurveda medicine due to their biological anti-inflammatory, anti-diabetic, and anti-enteritic activities that ward off the symptoms of malaria and ameliorate heart disease [16]. Basil is called the “Mother Medicine of Nature” or “The Queen of Herbs” [17]. The main phytochemicals contained in basil leaves are phenolic acids, flavonoids, propenyl phenol, and terpenoids, particularly, ursolic acid, which is often used as a biomarker [18]. Ursolic acid exhibits anti-inflammatory, antioxidant, anti-apoptotic, and anti-carcinogenic effects [19].

While there are interesting and convincing studies promoting the consumption of plant materials to enhance human health through the bioactive compounds they contain, a systemic database of the bioactive molecules in Massaman and Pad-ka-proa has not been reported. Therefore, this research aimed to identify and characterize the phenolic compounds contained in Massaman and Pad-ka-proa by LC-ESI-QTOF-MS/MS. These results will be useful for the functional food industry, medicinal applications, and health-conscious customers.

## 2. Materials and Methods

### 2.1. Chemicals and Reagents

Folin–Ciocalteu’s reagent for total phenolic content (TPC) was purchased from Loba Chemie Pvt.Ltd., Mumbai, India.

2,2-diphenyl-1-picryl hydrazyl (DPPH), 2,2-azino-bis-3-ethylbenzthiazoline-6-sulfonic acid (ABTS), 2,4,6-tripyridyl-s-triazine (TPTZ) for FRAP and a fluorescein solution for ORAC were purchased from Sigma-Aldrich, Darmstadt, Germany. 

Trolox (standard for TPC, DPPH, ABTS, FRAP, and ORAC) and rutin (standard for TFC) were high-performance liquid chromatography HPLC-grade and purchased from Sigma-Aldrich, Darmstadt, Germany.

2,2′-azobis(2-methylpropionamidine) dihydrochloride or AAPH used in ORAC were purchased from FUJIFILM Wako Pure Chemical Corporation, Miyazaki, Japan.

Methanol, acetonitrile, and acetic acid were HPLC-grade and purchased from RCI Labscan, Bangkok, Thailand.

### 2.2. Preparation of Curry Powders

Massaman curry and spicy basil leaf curry were made by mixing the ingredients listed in Table 1. All ingredients were purchased from a local market in Hat Yai, Songkhla province, Thailand. 

After grading and washing with a 100 ppm chlorine solution at a ratio of 1:3 (ingredient: solution) for 15 min, the fresh raw materials were rinsed with tap water 2 times to remove excess chlorine residue to lower than 1 ppm. The cleaned ingredients of each curry recipe were blended to a paste before drying using a drum dryer (DD-D12L16, Chareontut, Samutprakarn, Thailand) at 110–120 °C for 2–3 min to obtain dried curry powder with a moisture content of 4–6%. The spicy basil leaf curry was dried in a rotary hot air oven (HS-169, AT Packing, Nonthaburi, Thailand) at 70 °C for 16–18 h to obtain a moisture content of 4–6%. Each dried sample was ground with a high-speed mixer (WF-20B, Thaigrinder, Thailand) until the powder size was lower than 60 mesh (250 µm) (Laboratory test sieve, Endecotts, UK). Flow charts showing an overview of making the Massaman and spicy basil leaf curry powders are presented in Figure 1 and Figure 2, respectively.

### 2.3. Total Phenolic Content and Antioxidant Activity

#### 2.3.1. Sample Preparation and Extraction

Each curry powder sample was extracted following the method described by Srisook et al. [20] with some modifications, including using 80% ethanol and 24 h instead of 95% and 5 days. All powders from each curry sample were extracted with 80% ethanol at a ratio of 1:10 (curry powder: 80% ethanol) and stirred in the dark at 25 °C for 24 h. The mixtures were then separated by vacuum suction using a Buchner funnel before centrifugation (CR22GIII, Hitachi, Japan) at 4 °C for 20 min at 7100× *g*. The ethanol was completely removed using an evaporator (N-1000, EYELA, Rikakikai, Japan) before freeze-drying (KD-330cr, I.T.C., Bangkok, Thailand) at −25 °C until reaching a moisture content of 6–8%.

#### 2.3.2. Total Phenolic Content (TPC) Determination

(TPC) was determined using the method described by Singleton and Rossi [21] with some modifications, including using a well plate instead of a test tube. Briefly, 20 µL of the sample extract was added to a 96-well plate, followed by 100 µL of 10% Folin reagent (*v*/*v*). After incubation in the dark at 30 °C for 6 min, 7.5% Na_2_CO_3_ (anhydrous) (*w*/*v*) was added, and the mixture was incubated for another 30 min. The absorbance was measured at 765 nm using a microplate reader (Varioskan LUX, Thermo Scientific, Singapore, Singapore). TPC was measured using gallic acid as the standard agent at concentrations of 0–100 µg/mL with R^2^ = 0.999. The standard curve is shown in Appendix A. 

#### 2.3.3. Total Flavonoid Content (TFC) Determination

(TFC) was determined using the method described by Chandra et al. [22] with some modifications, including using a well plate instead of a test tube. Briefly, 100 µL of the sample extract was mixed with 100 µL of 2% AlCl_3_·6H_2_O (*w*/*v*) and incubated in the dark at 30 °C for 60 min. The absorbance of the mixture was then measured at 420 nm by a microplate reader (Varioskan LUX, Thermo Scientific, Singapore) using rutin as the standard agent at concentrations of 0–80 µg/mL with R^2^ = 0.998. The standard curve is shown in Appendix A.

#### 2.3.4. DPPH Radical Scavenging Activity

2,2-diphenyl-1-picryl hydrazyl (DPPH) radical scavenging activity was determined following the method of Ding et al. [23]. First, 100 µL of the sample extract was mixed with 100 µL of 0.2 mM DPPH in 95% ethanol. The mixture was then incubated in the dark for 30 min at 30 °C. Finally, the absorbance was measured at 517 nm by a microplate reader (Varioskan LUX, Thermo Scientific, Singapore, Singapore) using Trolox as the standard agent at concentrations of 0–12 µg/mL with R^2^ = 0.998. The standard curve is shown in Appendix A.

#### 2.3.5. ABTS Radical Scavenging Activity

The 2,2-azino-bis-3-ethylbenzthiazoline-6-sulfonic acid (ABTS) assay was determined as described by Arnao et al. [24]. The ABTS radical was generated by incubating 7.4 mM ABTS solution in the dark at 30 °C for 12 h. The radical solution was then diluted to obtain an absorbance of 1.1 ± 0.02 at 734 nm. Then, 20 µL of the sample extract was mixed with 280 µL of the radical solution and kept in the dark for 2 h at 30 °C. The absorbance of the mixture was measured at 734 nm by a microplate reader (Varioskan LUX, Thermo Scientific, Singapore) using Trolox as the standard agent at concentrations of 0–110 µg/mL with R^2^ = 0.999. The standard curve is shown in Appendix A.

#### 2.3.6. Ferric Reducing Antioxidant Power (FRAP) Assay

The ferric-reducing antioxidant power (FRAP) assay was determined following the method of Benzie and Strain [25]. A freshly prepared FRAP solution containing 300 mM acetate buffer pH 3.6, 10 mM TPTZ (2, 4, 6-tripyridyl-s-triazine) in 40 mM HCl and 20 mM FeCl_3_·6H_2_O (ratio 10:1:1) was warmed at 37 °C for 30 min. Then, 15 µL of the sample extract was mixed with 285 µL of the FRAP solution and incubated for 30 min at 37 °C. The absorbance of the mixture was measured at 593 nm by a microplate reader (Varioskan LUX, Thermo Scientific, Singapore) using Trolox as the standard agent at concentrations of 0–100 µg/mL with R^2^ = 0.999. The standard curve is shown in Appendix A.

#### 2.3.7. Oxygen Radical Absorbance Capacity (ORAC) Determination

The ORAC (oxygen radical absorbance capacity) was determined following the method of Huang et al. [26]. A sample solution of 25 µL was mixed with 150 µL of fluorescein solution 81.6 nM. The mixture was incubated at 37 °C for 15 min and then 25 µL of AAPH 153 mM was added. Fluorescence (excitation wavelength at 485 nm and emission wavelength at 530 nm) was read at 2 min time intervals for 90 min by a microplate reader (Varioskan LUX, Thermo Scientific, Singapore) using Trolox as the standard agent at concentrations 0–170 µg/mL with R^2^ = 0.998. The standard curve is shown in Appendix A.

The antioxidation activities were previously studied [27], and the details are shown in Table 2.

### 2.4. Characterization of Phenolic Compound Profiles by LC-ESI-QTOF-MS/MS 

All extract samples were examined for phenolic and flavonoid compounds following the modified method of Araujo et al. [28] and profiling by LC-ESI-QTOF-MS/MS-positive and -negative electrospray ionization at the University Center Laboratory with ISO accreditation. Each sample (2 µL) was injected into a Zorbax Eclipse Plus C18 Rapid Resolution HD column (150 mm length × 2.1 mm inner diameter) and performed at 25 °C. The mobile phases were solvent A, a mixture of methanol: acetonitrile: water: acetic acid (10:5:85:1, *v*/*v*), and solvent B, a mixture of methanol: acetonitrile: acetic acid (60:40:1, *v*/*v*), with a flow rate of 0.2 mL/min. Wavelengths (λ) at 230, 257, 280, 325, 368, and 450 nm were used to detect the compounds in the sample. Mass spectrometry was run on a Dual AJS ESI for ion source with an MSQ-TOF (model: G6545A, Agilent, Beijing, China) and mass spectrometer range of 100–1500 *m*/*z*. Electrospray ionization (ESI) was performed when the gas temperature reached 325 °C with a flow rate of 13 L/min and a nebulizer pressure set at 35 psig for the introduction source. Data were analyzed by MassHunter WorkStation Software Quantitative Analysis Navigator V8 and WorkStation Software Qualitative Analysis Workflows V8 with database MassHunter METLIN PCD. An overview diagram of the characterization of the phenolic profiles by LC-ESI-QTOF-MS/MS is presented in Appendix A.

### 2.5. Sensory Evaluation

Massaman curry powder and spicy basil leaf curry powder were cooked with the ingredients shown in Table 3 and then presented to 50 panelists for sensory evaluation using a 9-point Hedonic scale following the method of Wichchukit and O’Hahony [29] with ethical approval no. PSU-HREC-2023-008-1-1. Appearance, color, odor, taste, texture, and overall liking attributes were scored. Plain rice was served with the curry dishes, and mouthfeel was rinsed using normal water and fresh cucumber. This study complied with the Declaration of Helsinki and was approved by the human research ethics committee of Prince of Songkla University (PSU-HREC-2023-008-1-1).

### 2.6. Statistical Analysis

The experiment was set up using a completely randomized design (CRD). All quality parameters were performed with eight repetitions. Differences in mean values and variations were tested using ANOVA with Tukey’s test (*p* < 0.05). Statistical analysis of the data was carried out using SPSS statistics software version 22 (IBM, New York, NY, USA).

## 3. Results and Discussion

### 3.1. Total Phenolic and Flavonoid Contents and Antioxidant Activities

Phenolic compounds are important phytochemical constituents showing redox properties responsible for antioxidant activity with diverse benefits in the human diet [30]. The results indicated that the TPC of spicy basil leaf curry was significantly 6-fold higher than Massaman curry (*p* < 0.05) (Table 2), while the TFC value of Massaman curry was similar to spicy basil leaf curry. Lu et al. [31] studied the antioxidant capacity and contained phenolic compounds of 18 spices in curry powder, including star anise, fennel, cumin, angelica dahurica root, green prickleyash, Sichuan pepper, dried tangerine peel, white pepper, nutmeg, galangal, dried ginger, tsaoko amomum fruit, villous amomum fruit, dried chili pepper, bay leaves, cinnamon, and mustard, and found that total flavonoids were higher than total phenolics. Total flavonoids were reported as major constituents found in cardamom, clove, cinnamon, black pepper, cumin seed, fennel seed, red chili, coriander, and ginger [32]. Akullo et al. [33] stated that garlic bulbs extracted with ethanol provided higher TPC than TFC, explaining the reason why spicy basil leaf curry contained higher TPC than Massaman curry, even when the spicy basil leaf curry was mainly garlic (45%) and dried basil leaves (35%). Chaudhary et al. [34] reported that using methanol as the solvent for basil leaf extraction provided higher TFC than TPC. Solvent type plays a key role in extraction due to polarity and leads to various phytochemicals.

In this study, antioxidant DPPH and ABTS activities were assessed using Trolox as a standard and determined mainly via hydrogen and electron transfer [35,36]. The FRAP assay was determined by the electron transfer ability of antioxidants by reducing the colorless complex ferric ion (Fe^3+^) to the blue ferrous complex (Fe^2+^) [36], while the ORAC assay measured the ability to transfer hydrogen atoms to RO•/ROO• radicals generated by AAPH thermolysis in the presence of a probe that quantified antioxidant oxidation [37]. The results showed that the spicy basil leaf curry expressed higher antioxidant activity values than Massaman curry in all assays (*p* < 0.05) (Table 2). Dat-arun et al. [8] reported that fresh Massaman curry paste provided DPPH with 11.81 ± 0.06 mg GAE/100 g crude extract and FRAP with 0.311 ± 0.006 mg TE/100 g crude extract. However, to date, no scientific information on spicy basil leaf curry is available. Juntachote and Berghofer [38] found that basil leaves (*Ocimum sanctum* Linn) recorded DPPH with IC_50_ 20.6 µg extract/mL. Pearson’s correlations of the TPC, TFC, DPPH, ABTS, FRAP, and ORAC assays were significantly correlated at *p* < 0.01 with r > 0.974, while TFC was significantly correlated with ORAC at *p* < 0.05 with r = 0.898, as shown in Appendix A. This result suggests that the phytochemicals contained in both curries effectively inhibited the peroxyl radical generated in the human body [39]. Schaich et al. [40] found that DPPH, ABTS, and the ORAC assay had a good relationship and reacted with radicals through a similar mechanism with some modifications. For instance, antioxidants react with DPPH by transferring electrons and/or giving a hydrogen atom back to active molecules or radicals [41]. ABTS radicals were in an inactive form by taking electrons and hydrogen atoms from the antioxidant, while the ORAC assay evaluated the ability of an antioxidant to quench radicals by hydrogen atom transfers independently of electron transfers. The FRAP mechanism is based on electron transfer rather than hydrogen atom transfer [42]. 

### 3.2. Characterization of Polyphenols in Spicy Basil Leaf Curry and Massaman Curry Using LC-ESI-QTOF-MS/MS

Qualitative identification of the polyphenols in the spicy basil leaf and Massaman curries was conducted by LC-ESI-QTOF-MS/MS in both the negative and positive ionization modes (Table 4 and Table 5), and the contained phytochemicals and their biological activities are listed in Table 6. The major constituents found in spicy basil leaf curry were flavonoids and derivatives, comprising 17 compounds, 14 terpenes, 10 phenolics and derivatives, and 21 other types including quinones, alkaloids, chromones, capsaicinoids, flavonoidal alkaloids, and steroidal saponins (Table 4). The main flavonoid derivative was identified as 6-C-beta-D-Xylopyranosyl-8-C-alpha-L-arabinopyranosylapigenin (with abundance: 43.35 × 10^5^), which agreed with the finding of various herbs and spices [43]. Apigenin is a flavonoid compound that effectively downregulates the expression and secretion of pro-inflammatory cytokines through the IL-23/IL-17/IL-22 axes [44]. Apiin was the second-most abundant flavonoid (abundance: 40.55 × 10^5^), which is mainly found in celery leaves, parsley leaves, and bell peppers. In this experiment, apiin was found in fresh green and red chili, which are in the same genus as bell pepper (*Capsicum annuum* L.) [45]. Adem et al. [46] reported that apiin, hesperidin, rutin, and diosmin were the most effective agents against SARS-CoV-2 Mpro when compared with Nelfinavir (positive control). Cynaroside A (abundance: 31.11 × 10^5^) was found in coriander, basil, eggplant, and ginger rhizome [47]. Both cymaroside and luteolin-7-O-glucoside expressed multiactivity including anti-cancer, anti-bacterial, and antioxidant activity [48]. Song and Park [49] stated that luteolin and luteolin-7-O-glucoside increased the function of heme oxygenase-1, which exhibited a critical role in maintaining cellular redox homeostasis against oxidative stress. The highest terpene contents in the curry were capsianoside II (abundance: 52.23 × 10^5^) and capsianoside I (abundance: 52.23 × 10^5^), which were mainly liberated from chili (*Capsicum annuum* L.). The main ingredient used in spicy basil leaf curry was chili pods at 11%. Capsianosides, particularly capsianoside F, exhibited tight junction permeability of the human intestine, which mitigated leaky gut syndrome [50,51]. Dihydrocapsaicin (abundance: 18.81 × 10^5^) is commonly found in chili pods and diminishes TNFα-mediated activation of NFkB and its molecular targets in endothelial cells while also inducing upregulation of nitric oxide and exhibiting antioxidant properties [52]. Phenolics and derivatives such as N-feruloyltyramine (abundance: 63.50 × 10^5^) are commonly found in garlic bulbs [53] and mitigate several cardiovascular disorders through cyclooxygenase enzymes I and II [54]. These enzymes play a key role in P-selection, which mediates the formation of platelets and leukocytes in activated endothelial cells [54]. Quinic acid, a phenolic compound (abundance: 44.58 × 10^5^), is found in cinchona bark, coffee beans, tobacco leaves, carrot leaves, and apples. The National Center for Advancing Translational Sciences [55] reported multiple functions for quinic acid such as acting as an antioxidant that has shown anti-cancer activity through apoptosis-mediated cytotoxicity in breast cancer cell testing in mice models [56]. N-trans-Feruloyloctopamine (abundance: 7.79 × 10^5^) is mainly found in garlic and shows high potential as a tyrosinase inhibitor [57], relating to melanin production and skin cancer or carcinoma cells [58]. Extracted garlic skin containing N-trans-feruloyloctopamine inhibited cell proliferation and invasion in hepatocellular carcinoma cells [59]. 6′-Hydroxysimvastatin has been used as a cholesterol-lowering and anti-cardiovascular disease drug [60], while hydrocodone, also found in this study, is a morphinane-like compound commonly used in combination with acetaminophen to control moderate to severe pain [61]. The results showed that spicy basil leaf curry contained high amounts of phytochemicals with various health-promoting properties.

The Massaman curry in this experiment contained 54 flavonoids and derivatives, 23 phenolics and derivatives, 8 terpenes, and 13 other types including quinones, alkaloids, chromones, and ketones, as shown in Table 5. Apigenin 7-O-glucoside is a main flavonoid compound (abundance: 167.45 × 10^5^) generated from several plants [62]. *Candida* spp., which is the most common cause of yeast infection, is inhibited by Apigenin 7-O-glucoside, which also shows cytotoxic effects on colon cancer cells and cervical cancer HeLa cells as well as alleviating DSS-induced colitis [63]. Apigenin 7-O-glucoside significantly exhibited these mentioned activities [64,65,66]. Kaempferol 4′-glucoside was also found in Massaman curry at an abundance of 155.85 × 10^5^. It has various properties including anti-cancer, anti-inflammatory, antioxidant, anti-depressant, and anti-epilepsy properties, and it also improves cerebral blood flow [67]. Chang et al. [68] reported that kaempferol 4′-glucoside showed anti-inflammatory activity by inhibiting NO generation, iNOS protein, and iNOS mRNA level by retarding NF-κB-mediated iNOS gene transcription. Kaempferol also showed significant inhibition of NSCLC (non-small cell lung cancer) cell proliferation (*p* < 0.05) and inhibited the mesenchymal–epithelial transition in progressive lung cancer by promoting NSCLC cell autophagy, leading to NSCLC cell death in a rat model [69]. Yu et al. [70] stated that kaempferol reduced inflammatory bowel disease (IBD) by inhibiting IL-1β, IL-6, TNF-α, CRP, and NO secretion as well as retarding regenerated blood vessels of high intestinal microvascular density [71]. Luteolin (abundance: 91.64 × 10^5^) is a flavone type that is generally present in plants, with multiple functions such as antioxidant, anti-inflammatory, and antiallergic properties, in particular, against liver disorders, including metabolic-associated fatty liver disease, hepatic fibrosis, and hepatoma [72,73]. He et al. [74] reported that luteolin inhibited Aβ-induced oxidative stress, mitochondrial dysfunction, and neuronal apoptosis via a PPARγ-dependent mechanism, one of the pathways for Alzheimer’s disease in rat models. Wang et al. [75] stated that luteolin inhibited herpes simplex virus 1 (HSV-1) infection, enhanced antiviral type I interferon production, and activated the cytoplasmic DNA-sensing cGAS-stimulator of the interferon gene (STING) pathway. The main phenolic acid and derivative compound in Massaman curry was glucocaffeic acid (abundance: 52.54 × 10^5^), which is common in both herbs and spices [76,77]. The multifunctions of caffeic acid have been addressed as anti-cancer, antiviral, and anti-inflammatory activities [78]. Caffeic acid recovered ischemia-induced synaptic dysfunction in mouse hippocampal slices [79]. The results indicated that caffeic acid (1–10 μM) did not directly affect synaptic transmission and plasticity but indirectly affected other cellular targets to correct synaptic dysfunction. Quinic acid (abundance: 40.84 × 10^5^) and N-Feruloyltyramine (abundance: 30.35 × 10^5^) were also identified in spicy basil leaf curry. One terpene found in this experiment was cofaryloside (abundance: 16.11 × 10^5^). Cofaryloside I-II has been reported in Yunnan Arabica coffee beans [80,81,82], but no scientific information is currently available. Betulinic acid (abundance: 13.19 × 10^5^) was the second-most abundant terpene compound found in this experiment. Melo et al. [83] reported that betulinic acid is present in various plants. Betulinic acid (50 mg/mL in water) elevated the plasma hormone levels of insulin and leptin and decreased levels of ghrelin hormone in high-fat feed rats. Other biological activities of betulinic acid include anti-HIV, anti-inflammatory, and anti-cancer activities [84,85]. Maslinic acid (abundance: 9.60 × 10^5^) was also found in this experiment. Maslinic acid is commonly found in many types of plants and exhibits many health aspects such as hypoglycemic effects, anti-inflammatory effects, neuroprotective effects, antioxidant effects, and anti-tumor effects [86,87]. Cao et al. [88] found that maslinic acid administration favored probiotic bacterial growth in PD mice, which helped to increase striatal serotonin, 5-hydroxyindole acetic acid, and γ-aminobutyric acid levels, reduced levels of tumor necrosis factor-alpha and interleukin 1β in the substantia nigra pars compacta, and significantly prevented dopaminergic neuronal-related Parkinson’s disease in a rat model.

The LC-ESI-QTOF-MS/MS results indicated that Massaman curry was higher in the number of polyphenolic types and variety of flavonoids and derivatives compared with spicy basil leaf curry activity. However, spicy basil leaf curry provided higher antioxidant activity based on TPC, TFC, DPPH, ABTS, FRAP, and ORAC, possibility because cynaroside A from basil leaves has strong antioxidant effects, as reported by [49,89,90], where cynaside A exhibited good antioxidant activity with a lower IC_50_ than quercitrin, rutoside, and protocatechuic acid [89]. 

Using the LC-ESI-QTOF-MS/MS technique indicated the possibility of toxins from Massaman curry and spicy basil leaf curry as podophyllotoxin (abundance: 3.12 × 10^5^) and clitidine (abundance: 4.61 × 10^5^), respectively. Podophyllotoxin is mostly generated in the rhizome of *Podophyllum* species, which grow widely across the Himalayan and Western China regions. Physicians have attempted to use this toxin for external genital and perianal warts caused by the human papillomavirus (HPV). HPV can be an opportunistic infection (OI) of HIV [91,92]. Clitidine is created by the poisonous mushroom (*Clitocybe acromelalga*), and some molds contaminate dried herbs and spices as well as nuts [93,94,95]. Therefore, safety awareness is needed. Clitidine may not be as harmful as aflatoxins and ochratoxins, but its presence indicates that the drying and storage processes for curry powders must be considered and submitted to the Thai FDA for controlling and warning entrepreneurs, companies, and consumers as well as public sectors. Data on the thermal degradation of podophyllotoxin could be destroyed at 114–118 °C [96], and clitidine has no information. Therefore, preventive systems such as washing with proper detergents, for example, bi-sodium carbonate, acetic acid, ozone, and calcium hydroxide [97], drying conditions, and storage with high vacuum values as well as irradiation may need to be applied. 

This is the first report using LC-ESI-QTOF-MS/MS to confirm phytochemicals and non-volatile compounds contained in mixed herbs and spices or curries supporting the body and wellness. This finding generally suggests not only the potential health impact or bioactivity associated with the unique composition of polyphenolics and flavonoids in Massaman or spicy basil leaf curry but also suggests that toxicity due to plant and mold toxin contamination also needs to be considered and managed. In addition, the identified phytochemical profiling found in both curries provided great evidence to extend the intensity determination of specific biological compounds further to obtain more value-added applications including functional ingredients and food, nutraceuticals, and medicinal products. 

**Table 6 foods-13-00582-t006:** Major phytochemicals found in spicy basil leaf curry and Massaman curry, their biological properties, and their possible plant sources.

No.	Phytochemical	Biological Activity	Mechanism	Plant Source	Reference	Sample
**Phenolic acid and derivatives**						
1	N-Feruloyltyramine	-Antithrombotic.	-Inhibits cyclooxygenase enzymes I and II.	garlic (*Allium sativum*) *Lycium barbarum*	[53,54,98]	Spicy basil leaf curry and Massaman curry
		-Neurogenesis and neurotrophins.	-TrkA/ERK/CREB signaling pathway.			
2	Quinic acid	-Antioxidation.	-Inhibits hydrogen atom transfer, electron transfer, and sequential proton loss electron transfer activities.	cinchona bark, coffee beans, tobacco leaves, carrot leaves, apples, etc.	[55,56,99,100]	Spicy basil leaf curry and Massaman curry
		-Anti-cancer.	-Apoptosis-mediated cytotoxicity.			
		-Anti-inflammatory.	-Inhibits TNF-α-stimulation by inhibiting the MAP kinase and NF-κB signaling pathways.			
3	N-Feruloyltyramine	-Anti-cancer.	-Inhibits tyrosinase gene expression and melanine accumulation in melanoma cells. -Decreases the phosphorylation levels of Akt and p38 MAPK and EMT hepatocellular carcinoma cells.	garlic (*Allium sativum*) *Kali collinum* *Antidesma pentandrum* var. barbatum	[57,58,59]	Spicy basil leaf curry
4	Glucocaffeic acid	-Antioxidation.	-Inhibits hydrogen atom transfer and radical adduct formation activities.	various plants: coffee, fresh vegetables, fruits, tea, propolis, herbs, spices, etc.	[76,77,79,101,102,103]	Massaman curry
		-Neuroprotective.	-Affects synaptic transmission, plasticity, and dysfunction caused by oxygen–glucose deprivation (OGD).			
		-Anti-inflammatory.	-Inhibits the activity of NF-κB, IL-6, and STAT3 signaling.			
		-Antiviral.	-Inhibits the growth of both DNA and RNA viruses.			
		-Anti-cancer.	-Inhibits the proliferation of HeLa and ME-180 cells.			
**Flavonoids and derivatives**						
1	Apigenin	-Anti-inflammatory.	-Downregulates cytokines through the IL-23/IL-17/IL-22 axis.	herbs and spices	[43,44,104,105,106,107,108]	Spicy basil leaf curry and Massaman curry
		-Antibacterial.	-DNA gyrase harboring the quinolone-resistant S84L mutation.			
		-Anticancer.	-Inhibits the activity of the MAPK, PI3K/Akt, and NF-kB pathways.			
		-Antioxidation.	-Inhibits electron transfers and metal chelating activities.			
2	Apiin	-Antiviral.	-Against SARS-CoV-2 main protease.	various plants: celery leaves, parsley leaves, bell pepper, etc.	[45,46,109,110,111]	Spicy basil leaf curry and Massaman curry
		-Anti-inflammatory.	-Inhibits activity on nitrite (NO) and nitric oxide synthase (iNOS) expression.			
		-Anti-hypertension.	-Inhibits the activity of prostaglandin F_2α_ and angiotensin-I-converting enzyme.			
3	Cynaroside	-Anti-inflammatory.	-Inhibits the expression of iNOS, COX-2, TNF-α, and IL-6.	various plants: coriander, basil, eggplant, ginger, *Merremia tridentata* (L.), etc.	[48,112,113,114,115,116]	Spicy basil leaf curry and Massaman curry
		-Anti-diabetic.	-Strong α-amylase and α-glucosidase inhibitory activities.			
		-Antibacterial.	-Reduces the biofilm development of *Pseudomonas aeruginosa* and *Staphylococcus aureus* and reduces mutations leading to ciprofloxacin resistance in *Salmonella Typhimurium*.			
		-Antioxidation.	-Inhibits electron transfer and radical adduct formation activities.			
		-Anti-cancer.	-Decreases the phosphorylation level of AKT, mTOR, and P70S6K.			
4	Kaempferol 4′ glucoside	-Anti-inflammatory.	-Inhibits NO generation, iNOS protein, iNOS mRNA level, NF-κB, IL-1β, IL-6, IL-18, and TNF-α.	abundantly present in plants: tea, beans, broccoli, apples, herbs, etc.	[68,69,117,118,119]	Spicy basil leaf curry and Massaman curry
		-Anti-cancer.	-Inhibits NSCLC (non-small cell lung cancer) cell proliferation and promotes NSCLC cell autophagy and leading to NSCLC cell death.			
		-Neuroprotective.	-Inhibits Aβ deposition in Alzheimer’s disease and α-synuclein aggregation, Lewy body formation in Parkinson’s disease, and promotes dopamine release.			
		-Antioxidation.	-Inhibits electron transfer and hydrogen atom transfer.			
5	Luteolin	-Neuroprotective.	-Inhibits Aβ-induced oxidative stress, mitochondrial dysfunction, and neuronal apoptosis via the PPARγ-dependent mechanism.	abundantly present in plants: celery, parsley, broccoli, onion leaves, carrots, peppers, cabbages, apple, etc.	[120,121,122,123,124]	Massaman curry
		-Antioxidation.	-Hydrogen atom transfer and one electron transfer.			
		-Anti-inflammatory.	-Inhibits the activity of the MAPK and NF-kB pathways and SOCS3 in the signal transducer and activator of transcription 3 (STAT3) pathway.			
		-Anti-cancer.	-Luteolin strengthens tumor suppression of radiation and inhibits antiangiogenesis during radiation via decreased Integrin β1 expression.			
		-Anti-apoptotic.	-Reduces cleaved caspase-3 and Bax (pro-apoptotic factor) while increasing the Bcl-2 (antiapoptotic factor) signaling pathways.			
**Terpenes**						
1	Capsinoside	-Anti-leaky gut syndrome.	-Decreases G-actin and cytochalasin D and increases F-actin.	*Capsicum* plants	[50,125]	Spicy basil leaf curry
		-Antioxidation.	-Inhibits radical adduct formation activities.			
		-Anti-cancer.	-Inhibits homozygous mutations in PTEN and TP53 genes in the human prostate cancer cells line and inhibits mutation in codon 13 of the RAS proto-oncogene in colorectal carcinoma cells.			
2	Dihydrocapsaicin	-Anti-inflammatory.	-Inhibits TNF-α, NF-κB and nitric oxide.	*Capsicum* plants	[52,126,127]	Spicy basil leaf curry
		-Anti-cancer.	-Inhibits lysine-specific demethylase 1.			
		-Antioxidation.	-Inhibits electron-transfer activity.			
3	Cofaryloside	-Blood circulation enhancer.	-Found in plasma of rats after taking *Cyperi Rhizoma*, *Angelicae Sinensis Radix*, *Chuanxiong Rhizoma*, *Paeoniae Radix Alba*, and *Corydalis Rhizoma* but with no proven biological activities.	Yunnan Arabica coffee beans	[80,81,82,128]	Massaman curry
4	Betulinic acid	-Anti-obese.	-Increases the levels of insulin and leptin and decrease the level of ghrelin.	many fruits and vegetables	[83,129,130,131,132]	Massaman curry
		-Neuroprotective.	-Inhibits N- and T-type voltage-gated calcium channels.			
		-Anti-cancer.	-Inhibits the proliferation of liver cancer HUH7 and HCCLM3 cells by activating ferritinophagy in cancer cells and modulating the NCOA4/FTH1/LC3II signaling pathway for increase ferroptosis.			
		-Anti-inflammatory.	-Inhibits the mRNA expressions of pro-inflammatory cytokines interleukin-1β (IL-1β), IL-6 and NF-κB and increases IL-10.			
		-Antioxidation.	-Inhibits electron-transfer activity.			
5	Maslinic acid or crategolic acid or 2α, 3β-dihydroxyolean-12-en-28-oic acid	-Hypoglycemic effect.	-Increases the expression of Beclin1, ATG1, and Bcl-2 mRNA while decreasing the expression of TNF-α and IL-1β, caspase-3 and Bax mRNA.	various plants: olive, loquat leaves, red dates, eucalyptus, crape myrtle, sage, plantain, *Prunella* *vulgaris*, etc.	[86,87,88,133,134,135,136]	Massaman curry
		-Anti-inflammatory.	-Inhibits the activation of NLRP3 inflammasome, IL-6, IL-1β, TNF-α, and iNOS and the COX2, AKT/NF-κB, and MAPK signaling pathways.			
		-Neuroprotective.	-Increases striatal serotonin, 5-hydroxyindole acetic acid, and γ-aminobutyric acid levels in gut microbiota and inhibits neuroinflammation by reducing tumor necrosis factor alpha and interleukin 1β.			
		-Antioxidation.	-Inhibits electron transfer.			
		-Anti-tumor.	-Inhibits IL-6 expression, induces JAK and STAT3 phosphorylation, and down-regulates STAT3-mediated protein Bad, Bcl-2, and Bax expression to treat gastric cancer.			

### 3.3. Sensory Evaluation

The results showed that both curries contained various phytochemicals, in particular, flavonoids, simple phenolic acids, and terpenes. Some bitter and astringent-tasting compounds such as caffeic acid, (−)-epicatechin, and (+)-catechin were also found [137,138,139]. Purves et al. [137] reported that the bitter threshold of quinine was 0.008 mM for humans. A bitter taste can be due to toxins, but beers, wines, dark chocolate, and coffee are bitter tasting [140,141]. Noble [142] stated that small amounts of simple phenolic acids resulted in a bitter taste, while higher phenolic content provided an astringent taste when combined with sourness. However, unpleasant bitter and astringent tastes can be modified by masking with sweet, salty, and umami flavors [143]. For sensory preference, preclinical studies indicated that bitter substances may have potent effects that stimulate the secretion of gastrointestinal (GI) hormones and modulate gut motility via the activation of bitter taste receptors located in the GI tract [144]. The results in Table 7 show that both curries recorded high scores for all attributes (>7/9), with no mention of a bitter taste. The seasoning (sugar and salt) and other combinations (Table 3) such as oil, coconut milk, and chicken breast used in this experiment could be modified or reduced to mask any unpleasant tastes via various proposed mechanisms including bitterness-depressant substance absorption, inhibiting receptor sites, and reducing the intensity of bitter molecules as polarity effects. For example, [145,146] explained that spicy and anesthetic effects from clove oil can reduce the bitter taste in oral drugs. Brideau [147] stated that the bitter taste of chlorpheniramine maleate and phenylpropanolamine could be reduced by the combination of citric acid and sodium bicarbonate with certain fruit flavors such as lemon, orange, and cherry.

## 4. Conclusions

The Massaman and spicy basil leaf curries obtained from blending several herbs and spices were rich in phytochemicals, especially phenolic and flavonoid compounds. Spicy basil leaf curry contained significantly higher phenolic and flavonoid compounds and antioxidant activity (DPPH, ABTS, FRAP, and ORAC) than Massaman curry. The characterization of polyphenols in Massaman curry using LC-ESI-QTOF-MS/MS indicated 23 phenolic acids and derivatives, 54 flavonoids and derivatives, 8 terpenes, and 14 other types, while spicy basil leaf curry was composed of 10 phenolic acids and derivatives, 17 flavonoids and derivatives, 14 terpenes, and 21 other types. Cynaroside or luteolin-7-O-glucoside from basil leaves had a strong antioxidant effect and was the major cause of high antioxidant activity in spicy basil leaf curry. Sensory evaluations of the Massaman and spicy basil leaf curries gave 100% acceptance, with good scores for all attributes. The phenolic and flavonoid compounds contained in both curries such as caffeic acid, (−)-epicatechin, and (+)-catechin induced a bitter and astringent taste, but the panelists did not comment on this because the seasoning and other combinations modified, reduced, and masked the unpleasant bitter taste with a good mouthfeel and overall sensation. However, the high savory perception and health benefits derived from phytochemicals require further study in animal and clinical trials.

## Figures and Tables

**Figure 1 foods-13-00582-f001:**
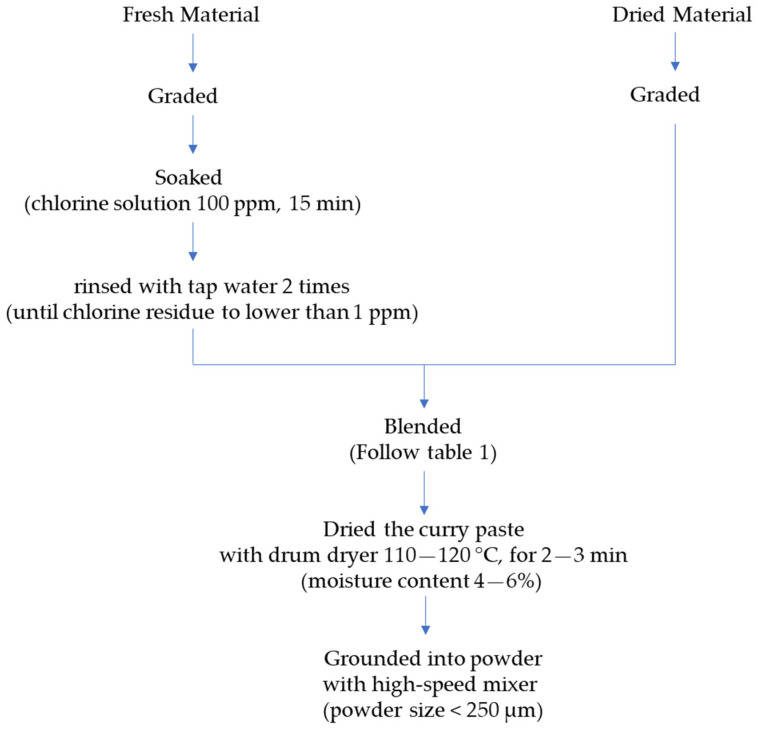
Flow chart showing the preparation of Massaman curry powder.

**Figure 2 foods-13-00582-f002:**
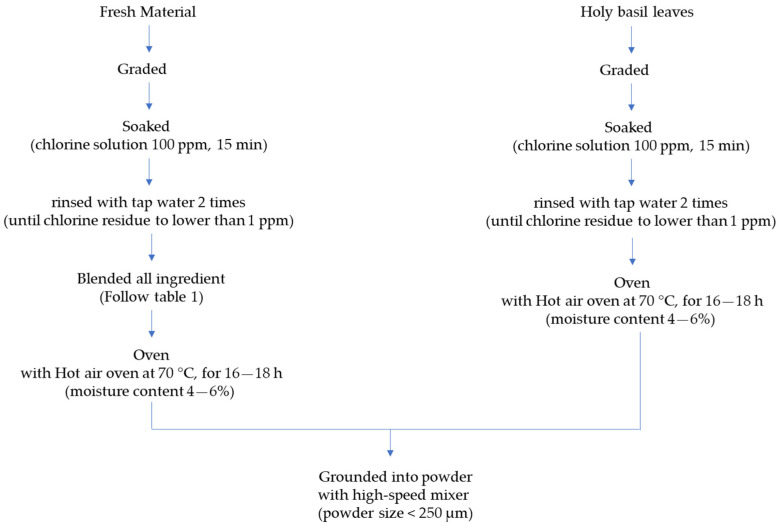
Flow chart showing the preparation of spicy basil leaf curry powder.

**Table 1 foods-13-00582-t001:** Ingredient ratios used in Massaman and spicy basil leaf curry powders.

Item	Ingredient	Harvest Time after Plantation or Flowering	Amount (%)	
			Massaman Curry	Spicy Basil Leaf Curry
1	fresh lemongrass (*Cymbopogon citrates* L.)	8–12 mo ^1^	5	-
2	fresh green chili (*Capsicum annuum* L.)	4–5 mo ^2^	-	5.5
3	fresh galangal (*Alpinia galanga* L.)	8–12 mo ^1^	5	-
4	fresh shallot bulb (*Allium ascalonicum* L.)	8–12 mo ^1^	35	-
5	fresh garlic bulb (*Allium sativum* L.)	8–12 mo ^1^	15	45
6	dried chili pepper (*Capsicum annuum* L.)	4–5 mo ^2^	15	-
7	dried black pepper (*Piper nigrum* L.)	4–5 mo ^2^	1	7
8	fresh ginger (*Zingiber officinale Roscoe.*)	8–12 mo ^1^	4	-
9	fresh coriander root (*Coriandrum sativum* L.)	6–8 mo ^1^	-	3
10	fresh red chili (*Capsicum annuum* L.)	4–5 mo ^2^	-	5.5
11	mixed spices *	No information	20	-
12	dried holy basil leaves (*Ocimum sanctum* L.)	4–6 mo ^1^	-	34

Mixed spices * include Kaffir lime skin, coriander seeds, caraway seeds, cloves, nutmeg seeds, cinnamon, cardamom, and nutmeg mace. ^1^ means harvest time after plantation. ^2^ means harvest time after flowering.

**Table 2 foods-13-00582-t002:** Total phenolic and flavonoid contents and antioxidant activity in Massaman curry and spicy basil leaf curry.

Antioxidant Activities	Massaman Curry	Spicy Basil Leaf Curry
TPC (µg GAE/g DW)	880.08 ± 48.46 ^b^	5595.29 ± 332.20 ^a^
TFC (µg RE/g DW)	271.36 ± 8.97 ^b^	371.26 ± 13.66 ^a^
DPPH (µg TE/g DW)	163.50 ± 1.79 ^b^	268.22 ± 3.01 ^a^
ABTS (µg TE/g DW)	2164.38 ± 2.04 ^b^	7923.68 ± 515.36 ^a^
FRAP (µg TE/g DW)	2757.64 ± 69.30 ^b^	5662.43 ± 247.86 ^a^
ORAC (mg TE/g DW)	1340.28 ± 18.76 ^b^	1940.10 ± 118.34 ^a^

All data are shown as mean ± standard deviation (sd). Different superscripts (a and b) indicate significant differences (*p* < 0.05), TPC: total phenolic content, TFC: total flavonoid content, DPPH: 2,2-diphenyl-1-picryl hydrazyl radical scavenging activity, ABTS: 2,2-azino-bis-3-ethylbenzthiazoline-6-sulfonic acid assay, FRAP: ferric-reducing antioxidant power assay, ORAC: oxygen radical absorbance capacity, TE: Trolox equivalent, RE: rutin equivalent, and DW: dried weight.

**Table 3 foods-13-00582-t003:** Ingredients and cooking ratios of Massaman and spicy basil leaf curries.

Item	Ingredient	Amount (g)	
		Massaman	Spicy Basil Leaves
1	Chicken breast	250	250
2	Curry powder	10	10
3	Salt	4	10
4	Sugar	17	25
5	Coconut milk	250	0
6	Plain water	250	30
7	Potato	100	0
8	Cooking oil	0	5

**Table 4 foods-13-00582-t004:** Characterization of variously identified phytochemicals found in spicy basil leaf curry as determined by LC-ESI-QTOF-MS/MS.

No.	Proposed Compound	Molecular Formula	*m*/*z*	Molecular Weight	RT (min)	ms/ms Product Ions	Abundant (×10^5^)
**Phenolic acid and derivatives**							
1	Quinic acid	C_7_ H_12_ O_6_	191.06	192.06	1.95	191.06	44.58
2	trans-Cinnamic acid	C_9_ H_8_ O_2_	147.05	148.05	3.77	103.06	2.35
3	Chlorogenic acid	C_16_ H_18_ O_9_	353.09	354.10	8.94	191.06	5.54
4	trans-p-Coumaric acid 4-glucoside	C_15_ H_18_ O_8_	325.09	326.10	9.76	163.04	4.04
5	Isoferuloyl C1-glucuronide	C_16_ H_18_ O_10_	415.09	370.09	14.72	415.09	2.86
6	7-Hydroxy-4-methylphthalide O-[arabinosyl-(1->6)-glucoside]	C_20_ H_26_ O_12_	457.13	458.14	18.00	457.13	6.11
7	N-trans-Feruloyloctopamine	C_18_ H_19_ N O_5_	328.12	329.13	19.58	328.12	7.79
8	Moschamine	C_20_ H_20_ N_2_ O_4_	351.13	352.14	20.58	351.13	2.37
9	N-(p-Hydroxyphenyl)ethyl p-hydroxycinnamide	C_17_ H_17_ N O_3_	282.11	283.12	21.18	282.11	2.57
10	N-Feruloyltyramine	C_18_ H_19_ N O_4_	312.12	313.13	21.76	312.12	63.50
**Flavonoids and derivatives**							
11	Pyrocatechol	C_6_ H_6_ O_2_	109.03	110.04	5.30	109.03	1.79
12	Cynaroside A	C_21_ H_32_ O_10_	443.19	444.20	7.41	443.19	31.11
13	Vitexin 4′-O-galactoside	C_27_ H_30_ O_15_	593.15	594.16	13.49	593.15	3.98
14	Apiin	C_26_ H_28_ O_14_	563.14	564.15	14.68	563.14	40.55
15	Isorhamnetin 3-lactoside	C_28_ H_32_ O_17_	639.16	640.16	15.76	639.15	2.12
16	Kaempferol 3-rhamnoside-(1->2)-rhamnoside	C_27_ H_30_ O_14_	577.16	578.16	16.84	577.15	5.51
17	Quercetin 3-galactoside	C_21_ H_20_ O_12_	463.09	464.10	18.20	463.09	6.12
18	Isorhamnetin 3-O-beta-(6″-O-E-p-coumaroylglucoside)-7-O-beta-glucoside	C_37_ H_38_ O_19_	785.19	786.20	19.78	785.19	3.75
19	(3″-Apiosyl-6″-malonyl)astragalin	C_29_ H_30_ O_18_	665.13	666.14	20.14	621.14	3.34
20	Keyakinin B	C_22_ H_22_ O_12_	477.10	478.11	20.31	477.10	11.88
21	6-C-beta-D-Xylopyranosyl-8-C-alpha-L-arabinopyranosylapigenin	C_25_ H_26_ O_13_	533.13	534.14	20.96	269.04	43.35
22	Mirificin	C_26_ H_28_ O_13_	547.15	548.15	21.26	269.05	3.20
23	6-C-Methylkaempferol 3-glucoside	C_22_ H_22_ O_11_	461.11	462.12	22.36	461.11	3.50
24	(±)-Naringenin	C_15_ H_12_ O_5_	271.06	272.07	24.94	271.06	4.01
25	Apigenin	C_15_ H_10_ O_5_	269.05	270.05	26.78	269.04	3.11
26	7,3′,4′-Trihydroxy-3,8-dimethoxyflavone	C_17_ H_14_ O_7_	329.07	330.07	28.34	329.07	1.17
27	Curcumin	C_21_ H_20_ O_6_	367.12	368.13	33.08	134.04	2.93
**Quinone**							
28	Idebenone metabolite (Benzenebutanoic acid, 2-hydroxy-3,4-dimethoxy-6-methyl-5-(sulfooxy)-)	C_13_ H_18_ O_9_ S	349.06	350.07	6.80	349.06	20.13
29	1,2,6,8-Tetrahydroxy-3-methylanthraquinone 2-O-b-D-glucoside	C_21_ H_20_ O_11_	447.09	448.10	15.49	447.09	4.68
30	Isosalsolidine	C_12_ H_13_ N O_2_	248.09	203.09	16.82	248.09	3.75
31	1,2,6,8-Tetrahydroxy-3-methylanthraquinone 2-O-b-D-glucoside	C_21_ H_20_ O_11_	447.09	448.10	17.71	447.09	54.67
32	Embelin	C_17_ H_26_ O_4_	293.18	294.18	32.48	221.15	3.85
**Terpene**							
33	Cincassiol B	C_20_ H_32_ O_8_	445.21	400.21	12.40	385.19	2.44
34	Hallactone B	C_20_ H_24_ O_9_ S	439.11	440.11	17.04	439.11	3.82
35	Capsianoside I	C_32_ H_52_ O_14_	659.33	660.34	29.20	659.33	19.04
36	Capsianoside III	C_50_ H_84_ O_26_	1099.52	1100.52	29.22	1099.52	3.42
37	(−)-Fusicoplagin A	C_24_ H_38_ O_7_	497.28	438.26	31.12	497.28	6.68
38	Capsianoside II	C_50_ H_84_ O_25_	1083.52	1084.53	31.25	1083.52	52.23
39	Lyciumoside IV	C_38_ H_64_ O_16_	775.41	776.42	32.75	775.41	1.43
40	Capsianoside VI	C_44_ H_74_ O_20_	921.47	922.48	32.96	921.47	3.00
41	Capsianoside IV	C_32_ H_52_ O_13_	643.33	644.34	34.26	643.33	6.15
42	Dihydrocapsaicin	C_18_ H_29_ N O_3_	306.21	307.21	34.74	170.16	18.81
43	Capsianoside D	C_82_ H_134_ O_38_	862.42	1726.85	34.86	862.42	7.94
44	Lucidenic acid M	C_27_ H_42_ O_6_	507.29	462.30	35.36	507.29	0.90
45	Cyclopassifloside VII	C_37_ H_62_ O_13_	759.42	714.42	35.62	759.42	1.11
46	Capsianoside F	C_82_ H_134_ O_37_	854.42	1710.86	36.18	854.42	16.35
**Alkaloid**							
47	(2E)-Piperamide-C5:1	C_16_ H_19_ N O_3_	272.13	273.14	25.38	272.13	20.63
48	Coumaperine	C_16_ H_19_ N O_2_	256.13	257.14	28.11	256.13	5.23
49	Feruperine	C_17_ H_21_ N O_3_	286.14	287.15	28.35	286.14	3.05
50	Piperolactam A	C_16_ H_11_ N O_3_	264.07	265.07	28.54	249.04	5.65
**Chromones**							
51	3′-Deaminofusarochromanone	C_15_ H_19_ N O_4_	276.12	277.13	5.90	276.12	2.65
52	Eugenitol	C_11_ H_10_ O_4_	205.05	206.06	15.40	205.05	4.14
53	5,7,3′,4′-Tetrahydroxy-4-phenylcoumarin 5-O-apiosyl-(1->6)-glucoside	C_26_ H_28_ O_15_	579.14	580.14	17.45	579.14	87.61
54	Cofaryloside	C_26_ H_42_ O_10_	513.27	514.28	24.82	513.27	17.28
**Capsaicinoid**							
55	Capsaicin	C_18_ H_27_ N O_3_	304.19	305.20	32.93	168.14	26.69
56	Homocapsaicin	C_19_ H_29_ N O_3_	318.21	319.21	34.94	182.15	1.96
57	Homodihydrocapsaicin	C_19_ H_31_ N O_3_	320.22	321.23	36.54	184.17	3.56
**Flavonoidal alkaloid**							
58	Ficine	C_20_ H_19_ N O_4_	336.12	337.13	29.34	336.12	1.07
**Steroidal saponins**							
59	Cistocardin	C_51_ H_84_ O_24_	1125.53	1080.53	33.61	1125.53	1.06
**Other**							
60	Clitidine	C_11_ H_14_ N_2_ O_6_	269.08	270.09	4.52	58.03	4.61
61	Hydrocodone	C_18_ H_21_ N O_3_	298.14	299.15	28.65	298.14	3.98
62	6′-Hydroxysimvastatin	C_25_ H_38_ O_6_	433.26	434.27	36.92	433.26	2.12

**Table 5 foods-13-00582-t005:** Characterization of variously identified phytochemicals found in Massaman curry as determined by LC-ESI-QTOF-MS/MS.

No.	Proposed Compound	Molecular Formula	*m*/*z*	Molecular Weight	RT (min)	ms/ms Product Ions	Abundant (×10^5^)
**Phenolic acid and derivatives**							
1	Quinic acid	C_7_ H_12_ O_6_	191.06	192.06	1.95	191.06	40.84
2	Shikimic acid	C_7_ H_10_ O_5_	173.05	174.05	2.01	93.03	4.72
3	Itaconic acid	C_5_ H_6_ O_4_	129.02	130.03	2.23	85.03	1.48
4	4-Glucogallic acid	C_13_ H_16_ O_10_	331.07	332.07	2.94	331.07	1.16
5	Gallic acid	C_7_ H_6_ O_5_	169.01	170.02	3.11	125.02	7.21
6	2-Hydroxycinnamic acid	C_9_ H_8_ O_3_	163.04	164.05	5.60	119.05	2.20
7	trans-p-Coumaric acid 4-glucoside	C_15_ H_18_ O_8_	325.09	326.10	5.62	163.04	8.43
8	Glucocaffeic acid	C_15_ H_18_ O_9_	341.09	342.10	6.55	341.09	52.54
9	5Z-Caffeoylquinic acid	C_16_ H_18_ O_9_	353.09	354.10	8.41	353.09	6.36
10	Chlorogenic acid	C_16_ H_18_ O_9_	353.09	354.10	9.04	191.06	4.35
11	Esculetin	C_9_ H_6_ O_4_	177.02	178.03	10.77	177.02	3.35
12	Dihydroconiferin	C_16_ H_24_ O_8_	343.14	344.15	13.03	59.01	23.97
13	Dihydromelilotoside	C_15_ H_20_ O_8_	327.11	328.12	13.20	165.06	3.32
14	3-O-Caffeoyl-4-O-methylquinic acid	C_17_ H_20_ O_9_	367.10	368.11	13.63	191.06	5.34
15	Dihydroferulic acid 4-O-glucuronide	C_16_ H_20_ O_10_	371.10	372.11	13.83	371.10	2.10
16	4-Feruloyl-1,5-quinolactone	C_17_ H_18_ O_8_	395.10	350.10	16.09	395.10	4.12
17	N-trans-Feruloyloctopamine	C_18_ H_19_ N O_5_	328.12	329.13	17.52	310.11	2.68
18	3-Hydroxychavicol 1-glucoside	C_15_ H_20_ O_7_	311.11	312.12	18.90	149.06	1.02
19	N-(p-Hydroxyphenyl)ethyl p-hydroxycinnamide	C_17_ H_17_ N O_3_	282.11	283.12	21.16	282.11	2.84
20	N-Feruloyltyramine	C_18_ H_19_ N O_4_	312.12	313.13	21.71	312.12	30.35
21	Orthothymotinic acid	C_11_ H_14_ O_3_	193.09	194.09	22.92	193.09	0.72
22	trans-Cinnamic acid	C_9_ H_8_ O_2_	147.05	148.05	23.47	103.06	0.70
23	2,8-Di-O-methylellagic acid	C_16_ H_10_ O_8_	329.03	330.04	25.17	329.03	3.57
**Flavonoids and derivatives**							
24	Procyanidin B2	C_30_ H_26_ O_12_	577.13	578.14	5.13	577.13	5.37
25	Cynaroside A	C_21_ H_32_ O_10_	443.19	444.20	7.41	443.19	4.84
26	(±)-Catechin	C_15_ H_14_ O_6_	289.07	290.08	8.16	289.07	5.98
27	Pavetannin B2	C_45_ H_36_ O_18_	863.18	864.19	10.53	863.18	13.99
28	(+)-Epicatechin	C_15_ H_14_ O_6_	289.07	290.08	12.48	289.07	7.77
29	Macrocarposide	C_21_ H_22_ O_11_	449.11	450.12	12.65	449.11	4.27
30	Vitexin 4′-O-galactoside	C_27_ H_30_ O_15_	593.15	594.16	13.49	593.15	15.15
31	Apiin	C_26_ H_28_ O_14_	563.14	564.15	14.68	563.14	74.47
32	Rutin	C_27_ H_30_ O_16_	609.14	610.15	15.16	609.14	14.92
33	Luteolin 4′-glucoside 7-galacturonide	C_27_ H_28_ O_17_	623.12	624.13	15.44	285.04	9.22
34	Isoorientin 3′-O-glucuronide	C_27_ H_28_ O_17_	311.06	624.13	15.49	311.06	8.10
35	Isovitexin	C_21_ H_20_ O_10_	431.10	432.11	16.89	431.10	2.33
36	Kaempferol 3-rhamnoside-(1->2)-rhamnoside	C_27_ H_30_ O_14_	577.16	578.16	16.94	577.16	2.39
37	6″-(4-Carboxy-3-hydroxy-3-methylbutanoyl) hyperin	C_27_ H_28_ O_16_	607.13	608.14	17.09	269.05	12.65
38	Kaempferol 4′-glucoside	C_21_ H_20_ O_11_	447.09	448.10	17.65	447.09	155.85
39	Quercetin 3′-O-glucuronide	C_21_ H_18_ O_13_	477.07	478.07	17.97	301.04	14.12
40	Quercetin 3-galactoside	C_21_ H_20_ O_12_	463.09	464.10	18.06	463.09	19.65
41	Tricetin 3′-xyloside	C_20_ H_18_ O_11_	433.08	434.08	18.67	433.08	13.52
42	Apigenin 7-O-glucoside	C_21_ H_20_ O_10_	431.10	432.11	19.65	431.10	167.45
43	Gossypetin 7-rhamnoside	C_21_ H_20_ O_12_	463.09	464.10	19.87	301.04	37.72
44	Quercetin	C_15_ H_10_ O_7_	301.04	302.04	19.90	301.04	45.78
45	Isoscoparin 2″-O-glucoside	C_28_ H_32_ O_16_	623.16	624.17	20.12	623.16	17.25
46	Eugenol O-[a-L-Arabinofuranosyl-(1->6)-b-D-glucopyranoside]	C_21_ H_30_ O_11_	517.19	458.18	20.28	293.09	3.66
47	Keyakinin B	C_22_ H_22_ O_12_	477.10	478.11	20.31	477.10	43.54
48	Orobol 8-C-(6″-acetylglucoside)	C_23_ H_22_ O_12_	489.10	490.11	20.42	489.10	30.15
49	6-C-beta-D-Xylopyranosylluteolin	C_20_ H_18_ O_10_	417.08	418.09	20.62	417.08	4.56
50	Vitexin 6″-(3-hydroxy-3-methylglutarate)	C_27_ H_28_ O_14_	575.14	576.15	21.03	575.14	5.64
51	Isorhamnetin 3-O-[4-Hydroxy-E-cinnamoyl-(->6)-b-D-glucopyranosyl-(1->2)-a-L-rhamnopyranoside]	C_37_ H_38_ O_18_	769.20	770.20	21.32	769.20	5.47
52	Hieracin	C_15_ H_10_ O_7_	301.04	302.04	21.69	301.03	2.30
53	Okanin 3,4-dimethyl ehter 4′-glucoside	C_23_ H_26_ O_11_	477.14	478.15	21.99	477.14	9.84
54	6-C-beta-D-Galactosylapigenin	C_21_ H_20_ O_10_	431.10	432.11	22.18	431.10	16.50
55	Fujikinetin 7-O-glucoside	C_23_ H_22_ O_11_	473.11	474.12	22.36	473.11	52.49
56	Apigenin	C_15_ H_10_ O_5_	269.05	270.05	22.40	269.05	89.56
57	3′,4′-Methylenedioxy epicatechin 5,7-dimethyl ether	C_18_ H_18_ O_6_	329.10	330.11	23.12	135.04	1.46
58	Villol	C_23_ H_22_ O_9_	441.12	442.13	23.83	279.07	9.06
59	Luteolin	C_15_ H_10_ O_6_	285.04	286.05	24.37	285.04	91.64
60	6″-O-Acetylvicenin 1	C_28_ H_30_ O_15_	605.15	606.16	24.60	605.15	1.91
61	(±)-Naringenin	C_15_ H_12_ O_5_	271.06	272.07	24.95	271.06	6.74
62	Iristectorigenin A 7-O-glucoside	C_23_ H_24_ O_12_	491.12	492.13	26.23	491.12	0.83
63	Licofuranocoumarin	C_21_ H_20_ O_7_	383.11	384.12	26.78	383.11	2.78
64	Diosmetin	C_16_ H_12_ O_6_	299.06	300.06	27.21	300.03	10.22
65	Isorhamnetin	C_16_ H_12_ O_7_	315.05	316.06	27.28	315.05	15.18
66	Apigenin 7-(6″-crotonylglucoside)	C_25_ H_24_ O_11_	499.12	500.13	27.78	499.12	3.10
67	Eugenol	C_10_ H_12_ O_2_	163.08	164.08	28.26	163.08	2.21
68	Formononetin	C_16_ H_12_ O_4_	267.07	268.07	28.81	267.07	1.60
69	5,3′,4′-Trihydroxy-3-methoxy-6,7-methylenedioxyflavone	C_17_ H_12_ O_8_	343.05	344.05	28.89	343.05	18.51
70	Procyanidin B1	C_30_ H_26_ O_12_	577.13	578.14	29.42	577.13	1.89
71	Prunetin	C_16_ H_12_ O_5_	283.06	284.07	31.76	283.06	4.93
72	Ovalitenin A	C_18_ H_14_ O_3_	277.09	278.09	32.23	277.09	1.10
73	Tetrahydrogambogic acid	C_38_ H_48_ O_8_	631.33	632.33	33.08	315.16	7.24
74	Curcumin	C_21_ H_20_ O_6_	367.12	368.13	33.16	134.04	0.94
75	2″,3″-Di-O-p-coumaroylafzelin	C_39_ H_32_ O_14_	723.17	724.18	33.51	723.17	6.07
76	Kaempferol 3-(2″,4″-di-(Z)-p-coumaroylrhamnoside)	C_39_ H_32_ O_14_	723.17	724.18	34.23	723.17	2.34
77	Broussinol	C_20_ H_22_ O_4_	325.14	326.15	36.42	325.14	38.90
**Quinone**							
78	Idebenone metabolite (QS-6)	C_15_ H_20_ O_6_	295.12	296.13	13.35	295.12	1.51
79	Isosalsolidine	C_12_ H_13_ N O_2_	248.09	203.09	16.82	248.09	7.46
80	Annocherine B	C_18_ H_17_ N O_4_	310.11	311.12	17.47	310.11	2.37
81	1,2,6,8-Tetrahydroxy-3-methylanthraquinone 2-O-b-D-glucoside	C_21_ H_20_ O_11_	447.09	448.10	19.43	447.09	11.38
82	13-Dihydroadriamycinone (Adriamycinol aglycone)	C_21_ H_20_ O_9_	415.10	416.11	22.77	415.10	4.49
83	7-Deoxy-13-dihydroadriamycinone	C_21_ H_20_ O_8_	399.11	400.12	24.74	399.11	4.96
84	Embelin	C_17_ H_26_ O_4_	293.18	294.18	32.48	236.10	2.20
**Terpene**							
85	Viguiestenin	C_21_ H_28_ O_7_	391.18	392.18	20.48	391.18	2.12
86	Cofaryloside	C_26_ H_42_ O_10_	513.27	514.28	24.85	513.27	16.11
87	Pseudolaric acid B	C_23_ H_28_ O_8_	431.17	432.18	27.36	431.17	9.20
88	Lucidenic acid M	C_27_ H_42_ O_6_	507.30	462.30	35.34	507.29	3.70
89	bayogenin	C_30_ H_48_ O_5_	487.34	488.35	35.56	487.34	1.68
90	Maslinic acid	C_30_ H_48_ O_4_	471.35	472.35	41.34	471.35	9.60
91	cis-p-Coumaroylcorosolic acid	C_39_ H_54_ O_6_	617.38	618.39	43.65	617.38	4.41
92	Betulinic acid	C_30_ H_48_ O_3_	455.35	456.36	44.61	455.35	13.19
**Alkaloid**							
93	Piperic acid	C_12_ H_10_ O_4_	263.06	218.06	10.37	263.06	1.24
94	(2E)-Piperamide-C5:1	C_16_ H_19_ N O_3_	272.13	273.14	25.34	272.13	5.11
**Chromones**							
95	Eugenitol	C_11_ H_10_ O_4_	205.05	206.06	15.39	93.03	0.74
96	3′-Deaminofusarochromanone	C_15_ H_19_ N O_4_	276.12	277.13	25.78	276.12	1.67
**Ketone**							
97	6-Gingerol	C_17_ H_26_ O_4_	293.18	294.18	30.80	57.03	1.57
98	(±)10-Gingerol	C_21_ H_34_ O_4_	349.24	350.25	38.48	57.03	4.20
**Other**							
99	Podophyllotoxin	C_18_ H_21_ N O_3_	298.14	299.15	28.65	298.14	3.98

**Table 7 foods-13-00582-t007:** Sensory scores of dishes cooked with Massaman curry and spicy basil leaf curry.

Item	Characteristic	Massaman Curry	Spicy Basil Leaf Curry
1	Appearance	7.40 ± 0.89 ^ns^	7.73 ± 0.87 ^ns^
2	Color	7.47 ± 0.90 ^ns^	7.43 ± 0.86 ^ns^
3	Odor	7.50 ± 1.04 ^ns^	7.60 ± 0.86 ^ns^
4	Flavor	7.50 ± 1.04 ^ns^	7.53 ± 0.78 ^ns^
5	Taste	7.57 ± 0.90 ^ns^	7.33 ± 1.52 ^ns^
6	Texture	7.60 ± 0.89 ^ns^	7.70 ± 0.65 ^ns^
7	Overall	7.60 ± 0.81 ^ns^	7.80 ± 0.71 ^ns^
8	Acceptance (%)	100	100
9	Non-acceptance (%)	0	0

All data are mean ± standard deviation (sd); ^ns^ means nonsignificant differences (*p* < 0.05).

## Data Availability

All data are contained within this article and the Appendix A.

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
