# Peer review of "Phytochemical Profiling and Antioxidant Activities of the Most Favored Ready-to-Use Thai Curries, Pad-Ka-Proa (Spicy Basil Leaves) and Massaman"

_foods, 2024, doi:10.3390/foods13040582_

Round 1

Reviewer 1 Report

Comments and Suggestions for Authors

Dear Author,

please find my review for the submitted manuscript:

The introduction contains a lot of information, but it could be improved in a few areas:
1. Clarity and Focus: The introduction jumps between discussing Thai curry dishes, the health benefits of herbs and spices, the impact of pollution on human health, the global reputation of Thai cuisine, and the potential benefits of phenolic compounds. It lacks a clear, focused topic or thesis statement, making it difficult for the reader to understand the main point.
2. Organization: The introduction lacks a clear structure and organization. It would benefit from breaking down the information into separate paragraphs with clear transitions between each topic.

3. Repetition: Some information about the health benefits of herbs and spices is repeated multiple times, which could be streamlined for clearer and more concise writing.

The materials and methods section would benefit from the following improvements:

1. Clarity and Precision: The section contains a lot of technical details, which can make it difficult for the reader to follow. It may be helpful to break down the information into smaller, more focused paragraphs with clear headings or subheadings for each procedure (e.g., Chemicals and Reagents, Preparation of fresh curry paste, Total phenolic content and antioxidant activity). This will help the reader to navigate the information more easily.
2. Consistent Format: The section could benefit from a consistent format for presenting the methods. For example, consistently using a step-by-step approach or bullet points for each procedure can make the information easier to digest and follow.

3. Use of Jargon: The section includes technical terms and abbreviations such as TPC, TFC, FBS, RPMI, etc. While it's important to provide technical details, it's also important to consider the audience and ensure that the terminology is explained or defined where necessary for non-expert readers.

The results section would benefit from the following improvements:

1.      The results section provides a comprehensive analysis of the polyphenolic composition, antioxidant activities, presence of toxins, and sensory evaluation of Massaman and spicy basil curry. The section begins by highlighting that the polyphenolics found in Massaman curry were higher in quantity and variety of flavonoids and derivatives compared to spicy basil curry. This finding suggests potential health benefits or bioactivity associated with the unique composition of polyphenolics in Massaman curry. However, it is important to integrate this information with further details on the potential health implications or bioactive properties associated with the identified polyphenolics.

2.      Furthermore, while the LC-ESI-QTOF-MS/MS technique indicated the possibility of toxins from both curries, including podophyllotoxin and clitidine, there is a need to expand on the implications for food safety and potential health risks associated with these toxins. Additionally, recommendations for proper handling, storage, or processing of curry powders should be included to address these safety concerns.

3.      In addition, the assessment of antioxidant activities based on TPC, TFC, DPPH, ABTS, FRAP, and ORAC revealed that spicy basil leaves curry provided higher antioxidant activity. This observation was attributed to the strong antioxidant effects of cynaroside A from basil leaves. It would be beneficial to include detailed references supporting the antioxidant activities of specific compounds to enhance the validity and significance of the reported findings.

4.      Furthermore, the sensory evaluation results indicated that both curries contained various phytochemicals, including flavonoids, simple phenolic acids, and terpenes. Additionally, bitter and astringent-tasting compounds such as caffeic acid, (−)-epicatechin, and (+)-catechin were identified. To provide a more comprehensive analysis of the sensory evaluation, it would be valuable to elaborate on how these compounds may impact the overall sensory attributes of the curries and discuss potential modifications in seasoning or combinations to enhance their palatability and consumer acceptance.

5.      Chemometrics methods could significantly support the findings in several ways. Firstly, chemometrics tools, such as multivariate data analysis and pattern recognition techniques, can be employed to extract meaningful information from the complex dataset generated by the analysis of polyphenolic composition, antioxidant activities, presence of toxins, and sensory evaluation of the curries. These methods can help in identifying patterns, trends, and relationships within the data that may not be readily apparent through univariate analysis.

Specifically, chemometrics techniques like principal component analysis (PCA) and partial least squares-discriminant analysis (PLS-DA) can be used to explore the overall variability in the polyphenolic profiles and antioxidants activities of the two curries. By employing these methods, it becomes possible to discern whether certain polyphenolic compounds or antioxidant activities contribute significantly to the differences observed between Massaman and spicy basil curry.

Moreover, chemometrics methods can aid in identifying potential biomarkers or characteristic chemical profiles associated with the different curry types. By applying chemometric models, it is feasible to pinpoint specific compounds or compound classes that are discriminative and influential in distinguishing between the curries. This information can provide valuable insights into the unique chemical fingerprints of each curry and elucidate the key compounds contributing to their distinct properties.

Additionally, chemometrics techniques can play a crucial role in addressing the presence of potentially harmful toxins in the curries. Specifically, methods like hierarchical cluster analysis or similarity analysis can assist in detecting similarities or dissimilarities in the toxin profiles of the curries, helping to identify common sources or potential contamination issues. Furthermore, chemometric models can aid in establishing predictive models for toxin detection, thus contributing to the development of strategies for ensuring the safety and quality of the curries.

Comments on the Quality of English Language

the language could be improved

Author Response

Reviewer 1 (Highlight with bule color)

Introduction

  1. Clarity and Focus: The introduction jumps between discussing Thai curry dishes, the health benefits of herbs and spices, the impact of pollution on human health, the global reputation of Thai cuisine, and the potential benefits of phenolic compounds. It lacks a clear, focused topic or thesis statement, making it difficult for the reader to understand the main point.
  • I already edited and rearranged it.
  1. Organization: The introduction lacks a clear structure and organization. It would benefit from breaking down the information into separate paragraphs with clear transitions between each topic.
  • I already edited and rearranged it.
  1. Repetition: Some information about the health benefits of herbs and spices is repeated multiple times, which could be streamlined for clearer and more concise writing.
  • I already edited and rearranged it.

Material and methods

  1. Clarity and Precision: The section contains a lot of technical details, which can make it difficult for the reader to follow. It may be helpful to break down the information into smaller, more focused paragraphs with clear headings or subheadings for each procedure (e.g., Chemicals and Reagents, Preparation of fresh curry paste, Total phenolic content and antioxidant activity). This will help the reader to navigate the information more easily.
  • I already edited.
  1. Consistent Format: The section could benefit from a consistent format for presenting the methods. For example, consistently using a step-by-step approach or bullet points for each procedure can make the information easier to digest and follow.
  • I already edited and added flow charts. (Figure 1, 2)

  1. Use of Jargon: The section includes technical terms and abbreviations such as TPC, TFC, FBS, RPMI, etc. While it's important to provide technical details, it's also important to consider the audience and ensure that the terminology is explained or defined where necessary for non-expert readers.
  • I already edited.

Result and discussion

  1. The results section provides a comprehensive analysis of the polyphenolic composition, antioxidant activities, presence of toxins, and sensory evaluation of Massaman and spicy basil curry. The section begins by highlighting that the polyphenolics found in Massaman curry were higher in quantity and variety of flavonoids and derivatives compared to spicy basil curry. This finding suggests potential health benefits or bioactivity associated with the unique composition of polyphenolics in Massaman curry. However, it is important to integrate this information with further details on the potential health implications or bioactive properties associated with the identified polyphenolics.
  • I already edited. (line 391-401)
  1. Furthermore, while the LC-ESI-QTOF-MS/MS technique indicated the possibility of toxins from both curries, including podophyllotoxin and clitidine, there is a need to expand on the implications for food safety and potential health risks associated with these toxins. Additionally, recommendations for proper handling, storage, or processing of curry powders should be included to address these safety concerns.
  • I already edited. (line 385-391)
  1. In addition, the assessment of antioxidant activities based on TPC, TFC, DPPH, ABTS, FRAP, and ORAC revealed that spicy basil leaves curry provided higher antioxidant activity. This observation was attributed to the strong antioxidant effects of cynaroside A from basil leaves. It would be beneficial to include detailed references supporting the antioxidant activities of specific compounds to enhance the validity and significance of the reported findings.
  • I already edited. (line 373-374)

  1. Furthermore, the sensory evaluation results indicated that both curries contained various phytochemicals, including flavonoids, simple phenolic acids, and terpenes. Additionally, bitter and astringent-tasting compounds such as caffeic acid, (−)-epicatechin, and (+)-catechin were identified. To provide a more comprehensive analysis of the sensory evaluation, it would be valuable to elaborate on how these compounds may impact the overall sensory attributes of the curries and discuss potential modifications in seasoning or combinations to enhance their palatability and consumer acceptance.
  • I already edited. (line 452-460)
  1. Chemometrics methods could significantly support the findings in several ways. Firstly, chemometrics tools, such as multivariate data analysis and pattern recognition techniques, can be employed to extract meaningful information from the complex dataset generated by the analysis of polyphenolic composition, antioxidant activities, presence of toxins, and sensory evaluation of the curries. These methods can help in identifying patterns, trends, and relationships within the data that may not be readily apparent through univariate analysis.
  • It was a great comment, however there was no supported program available in Faculty and time limited to run a PCA in this work. But I will keep your suggestion in my team for further work and analysis.

Reviewer 2 Report

Comments and Suggestions for Authors

I have read the submission entitled 'Phytochemical profiling and antioxidant activities of the most popular instant Thai curries, Pad-ka-proa (spicy basil leaves) and Massaman'. Curries are globally renowned dishes. It is a interesting report. The study is design good but too simple. Authors should consider the following suggestions before publication.

1. There is no conclusion in the Abstract section. Authors just simply compared two curries.

2. Text and grammar mistakes should be revised, such as In 2.2. Preparation of fresh curry paste, 'grad-ing'.

3. Why author rised the Table 3, is there any relationship with this study?

4. ppm value should be added in the Table 4.

Author Response

Reviewer 2 (Highlight with green color)

Introduction

  1. There is no conclusion in the Abstract section. Authors just simply compared two curries.
  • I already edited. (line 30-32)
  1. Text and grammar mistakes should be revised, such as In 2.2. Preparation of fresh curry paste, 'grad-ing'.
  • I already edited.
  1. Why author rised the Table 3, is there any relationship with this study?
  • Table 3 indicates the ingredients used in the recipe for sensory evaluation. I already made it clear that why suspected bitter compounds did not show any bitter taste problem because of combination of sugar, salt, coconut milk or oil and chicken breast.
  1. ppm value should be added in the Table 4.
  • In this study, LC-ESI-QTOF-MS/MS was used for qualitative not quantitative purposes. However, in the next step we plan to quantity the intensity of interested compounds. Therefore, ppm could not report at this time.

Reviewer 3 Report

Comments and Suggestions for Authors

In this study, Sunisa Siripongvutikorn and colleagues investigated the phytochemical profiling and antioxidant activities of the most popular instant Thai curries. The results of the study should provide original and useful information to the functional food industry, medicinal applications, and health-conscious customers. However, I have some concerns about the manuscript, below are the comments.

General comments:

1. Abstract section: This section lacks of some details of background. The authors should supplement the relevant background of this study and further to explain its significance.

2. Introduction section: The logic of the first paragraph is chaotic and needs to be rewritten. The author should focus on introducing the background related to the research topic and delete parts that are not relevant, such as line 40-43, which is not related to this study and is repeated in line 65-69.

3. Suggest merging this section (line 74-77) with the preceding paragraph.

4. The harvest time and location should be clarified in “Preparation of fresh curry paste” section.

5. Total phenolic content (TPC) was determined using Trolox as the standard agent? I think this method is unreasonable.

6. The standard curve of rutin shoud be added in line 133.

7. The standard curve of rutin shoud be added in line 139, line 148, line 157, and line 165.

8. The antioxidation activities were previously studied Table 2 was not necessary, it is recommended to delete it.

9. line 204: p was in italic type, whereas, p was not italic in other section ( such as line 211, line 232, line 237 and line 420), please unify the format in the entire manuscript.

10. Results and Discussion of total phenolic at line 208-224 shoud be re-written. Because there is few description about results, the discussion section should focus on the experimental results.

11. Where is the result of Pearson’s correlation (line 237-239)? The authors shoud add the related details of data.

12. The description of “The main flavonoid derivative was identified as 6-beta-D-Xylopyranosyl-8-C-alpha-L-arabinopyranosylapigenin” in line 259-259, is result or discussion? Why this sentence citing references [43]?

13. The writing of molecular formulas should be standardized. Such as, C7H12O6 is right, C7H12O6 should be corrected.

14. It is recommended to upload Table 6 as supporting material, as this is not a direct result of this study.

15. Could you merge the results of various compounds found in Massaman curry and spicy basil leaves into the same table? In my opinion, merging them into one table would facilitate readers to have a clearer understanding of the differences between Massaman curry and spicy basil leaves .

16. The reference format needs further improvement. Such as line 468 (Journal name), line 525 (Journal name), line 548 (pp) et al……. 

Author Response

Reviewer 3 (Highlight with yellow color)

Abstract

  1. Abstract section: This section lacks of some details of background. The authors should supplement the relevant background of this study and further to explain its significance.
  • I already edited. (line 19-21)

Introduction

  1. The logic of the first paragraph is chaotic and needs to be rewritten. The author should focus on introducing the background related to the research topic and delete parts that are not relevant, such as line 40-43, which is not related to this study and is repeated in line 65-69.
  • I already edited and rearranged it. (Highlight with bule color)
  1. Suggest merging this section (line 74-77) with the preceding paragraph.
  • I already edited and rearranged it. (Highlight with bule color)

Materials and Method

  1. The harvest time and location should be clarified in “Preparation of fresh curry paste” section.
  • I already edited and added flow charts. (Figure 1, 2) (Highlight with bule color)
  1. Total phenolic content (TPC) was determined using Trolox as the standard agent? I think this method is unreasonable.
  • Actually, standard agents for TPC and antioxidant activity can be Trolox, gallic acid and even vitamin C for checking functional group ability or to stimulate antioxidant in human body which vitamin E works well with vitamin C. Therefore, several papers used Trolox as standard agent to forecast antioxidant activity in human body. Since Trolox is used as an antioxidant standard agent, it also be used as agent for TPC to see its relationship when compared with other standards.

  1. The standard curve of rutin should be added in line 133, 139, 148, 157, 165.
  • I already edited and added standard curve. (Figure S1-S6)
  1. The antioxidation activities were previously studied Table 2 was not necessary, it is recommended to delete it.
  • My team and I already discussed this point before submitting the manuscript and we would like to use the Table 2 due to better explanation in other sections.
  1. Results and Discussion of total phenolic at line 208-224 should be re-written. Because there is few description about results, the discussion section should focus on the experimental results.
  • I already edited.
  1. Where is the result of Pearson’s correlation (line 237-239)? The authors should add the related details of data.
  • I already edited and added table S1 in supplementary data.
  1. line 204: p was in italic type, whereas, p was not italic in other section ( such as line 211, line 232, line 237 and line 420), please unify the format in the entire manuscript.
  • I already edited.
  1. The description of “The main flavonoid derivative was identified as 6-beta-D-Xylopyranosyl-8-C-alpha-L-arabinopyranosylapigenin” in line 259-259, is result or discussion? Why this sentence citing references [43]?
  • I already edited.
  1. The writing of molecular formulas should be standardized. Such as, C7H12O6 is right, C7H12O6 should be corrected.
  • I already edited.
  1. It is recommended to upload Table 6 as supporting material, as this is not a direct result of this study.
  • My team and I tried to do what you asked and found out that it lacks some essential information to compare with Table 4 and 5 therefore it should be there. In addition, other reviewers agreed to have it.

  1. Could you merge the results of various compounds found in Massaman curry and spicy basil leaves into the same table? In my opinion, merging them into one table would facilitate readers to have a clearer understanding of the differences between Massaman curry and spicy basil leaves.
  • My team and I tried to do it already, but it is difficult to read, understand, and follow, then could you please let us put it separately?
  1. The reference format needs further improvement. Such as line 468 (Journal name), line 525 (Journal name), line 548 (pp) et al.?
  • I already checked however; some reference is not manuscript but organization therefore it lacks of journal name. 

Round 2

Reviewer 1 Report

Comments and Suggestions for Authors

thnaks for the effor so far. Still, the chemmetrics is needed. Both R and other free online platforms can do the job

Author Response

Dear Reviewer 1,

Thank you for your suggestion. Due to the due date, and data analysis for chemometrics as you suggested and the explanation need more time to do and polish my work therefore, I worry that I cannot do the chemometrics as you suggested at this time, could you please accept my limitation.    

Best regarding

Sunisa and team

Reviewer 2 Report

Comments and Suggestions for Authors

All comments were addressed in this version.

Author Response

Dear Reviewer 2,

Thank you very much for your great support. 

Best regarding 

Sunisa and team 

Reviewer 3 Report

Comments and Suggestions for Authors

The author has made revisions according to my suggestions, however, I have some concerns about the manuscript, below are the comments. The author should carefully complete every revision and provided a detailed point-by-point response.

General comments:

1. line 25-26: There is an error in grammar, which needs to be rewritten.

2. I agree to the authors opnion that several study used Trolox as standard agent to antioxidant activity. Could use Trolox as the standard agent to determine Total phenolic content (TPC)? If you think it's feasible, can you list the relevant references? I still think this method is unreasonable.

3. The author's attitude towards revising the manuscript is not serious enough. For example, the format of Ref.6, 29, 39 was not rigorous, such as Journal name (LWT-FOOD SCI TECHNOL, Food Qual Prefer) and page number (pp). I have raised this issue before, but the author has not made any modifications.

Comments on the Quality of English Language

Quality of English Language need further improvment.

Author Response

Dear reviewer 3,

Thank you for your suggestions and sorry that I did not recheck the references as you mentioned properly. I would like to tell you that total phenolic content determination in my work includes gallic acid, gallic acid, ascorbic and Trolox as standards and has been published in the previous study (in Foods,  title "Antioxidant and nitric oxide inhibitory activity of the six most popular instant Thai curries, 2024).  However, I think it was better to use gallic acid as standard as you mentioned to not make any readers have doubted it.  I already edited it, and sorry to make you worry. Anyway, I also attached the reference using Trolox and gallic acid as standard for total phenolic content determination. For other points, please accept my error, however, I already edited (highlight with purple color). The edited manuscript will be sent to the editorial system because there is a limitation to submitting the review report system.     

Best regarding 

Sunisa and team  
